# Translocating Peptides of Biomedical Interest Obtained from the Spike (S) Glycoprotein of the SARS-CoV-2

**DOI:** 10.3390/membranes12060600

**Published:** 2022-06-10

**Authors:** Maria C. Henao, Camila Ocasion, Paola Ruiz Puentes, Cristina González-Melo, Valentina Quezada, Javier Cifuentes, Arnovis Yepes, Juan C. Burgos, Juan C. Cruz, Luis H. Reyes

**Affiliations:** 1Grupo de Diseño de Productos y Procesos (GDPP), Department of Chemical and Food Engineering, Universidad de los Andes, Bogota, DC 111711, Colombia; mc.henao10@uniandes.edu.co (M.C.H.); c.ocasion10@uniandes.edu.co (C.O.); 2Grupo de Investigación en Nanobiomateriales, Ingeniería Celular y Bioimpresión (GINIB), Department of Biomedical Engineering, Universidad de los Andes, Bogota, DC 111711, Colombia; p.ruiz@uniandes.edu.co (P.R.P.); c.gonzalez19@uniandes.edu.co (C.G.-M.); v.quezada@uniandes.edu.co (V.Q.); jf.cifuentes10@uniandes.edu.co (J.C.); 3Physical Sciences Program, Universidad de Cartagena, Cartagena 130015, Colombia; ayepesp1@unicartagena.edu.co; 4Grupo de Modelado Computacional (GruMoC), Chemical Engineering Program, Universidad de Cartagena, Cartagena 130015, Colombia

**Keywords:** spike glycoprotein, SARS-CoV-2, molecular dynamics, cell-penetrating peptides, drug delivery, biocompatibility

## Abstract

At the beginning of 2020, the pandemic caused by the SARS-CoV-2 virus led to the fast sequencing of its genome to facilitate molecular engineering strategies to control the pathogen’s spread. The spike (S) glycoprotein has been identified as the leading therapeutic agent due to its role in localizing the ACE2 receptor in the host’s pulmonary cell membrane, binding, and eventually infecting the cells. Due to the difficulty of delivering bioactive molecules to the intracellular space, we hypothesized that the S protein could serve as a source of membrane translocating peptides. AHB-1, AHB-2, and AHB-3 peptides were identified and analyzed on a membrane model of DPPC (dipalmitoylphosphatidylcholine) using molecular dynamics (MD) simulations. An umbrella sampling approach was used to quantify the energy barrier necessary to cross the boundary (13.2 to 34.9 kcal/mol), and a flat-bottom pulling helped to gain a deeper understanding of the membrane’s permeation dynamics. Our studies revealed that the novel peptide AHB-1 exhibited comparable penetration potential of already known potent cell-penetrating peptides (CPPs) such as TP2, Buforin II, and Frenatin 2.3s. Results were confirmed by in vitro analysis of the peptides conjugated to chitosan nanoparticles, demonstrating its ability to reach the cytosol and escape endosomes, while maintaining high biocompatibility levels according to standardized assays.

## 1. Introduction

Cell membranes are critical components of all living organisms and play a significant role as selective barriers to the entry of different solutes according to physiological needs. They are mainly composed of phospholipids, several types of embedded proteins (e.g., integral and peripheral membrane proteins [1]), and carbohydrates [2]. However, the phospholipid composition of cells from different tissues and the type of embedded proteins might vary significantly based on the required cellular functions, providing a suitable environment for an ample variety of biochemical reactions [3]. Besides its well-documented role of interface with the extracellular domain, the cell membrane has been reported to be involved in other cellular functions through multiple events occurring in different time scales, such as molecular transport, communication with the environment, transduction of signaling cascades, and control of various metabolic processes [2]. Cell membranes facilitate small and polar molecules’ intake to the intracellular domain, such as gases, ions, and amino acids. This process generally occurs by forming transient pores; however, this is not possible with larger size molecules [4]. As a result, several pharmacological molecules face challenges in crossing cell membranes, and only a tiny fraction (i.e., 1–10%) of what is administered to patients reaches the site of action. This limitation is exacerbated even further in the case of highly specialized physiological barriers such as the intestinal lumen and the blood-brain barrier (BBB) [5].

To overcome this major issue, drug design has mainly focused on targeting proteins such as enzymes and antibodies located in the extracellular space. However, these target proteins correspond to only one-third of the proteome, reducing the expected therapeutic action significantly in several conditions, including neurodegenerative, autoimmune, and oncogenic diseases [6]. This challenging situation complicates even further for treating several diseases whose therapeutic management relies heavily on intracellular targeting, such as Gaucher Disease [7], where blood cholesterol needs to be decreased by inhibiting the enzyme HMG-CoA reductase [8], or HIV, which can be mitigated by reverse transcriptase inhibitors that are only active intracellularly [9,10].

Currently, there are different methods to mediate the intake of cargoes to the intracellular space, including mechanical and electrical transfection techniques such as microinjection, ultrasonic nebulization, and electroporation [11]. Chemical and biochemical processes such as calcium phosphate co-precipitation, and viral carrier delivery systems have been tested successfully. In this case, some of the preferred choices include retroviruses, adenoviruses, and lentiviral vectors [12]. However, some of these methods have shown drawbacks in high cytotoxicity and immunogenicity and low delivery yields for the transported bioactive molecules [13]. Consequently, non-viral delivery methods such as cell-penetrating peptides (CPPs) have gained significant attention due to their ease of synthesis and functionalization, low cytotoxicity, and immunogenicity [14,15].

CPPs, also known as protein transduction domains (PTDs), are small peptides with lengths ranging from 5 to 30 amino acids [13]. They generally exhibit a positive net charge at a physiological pH due to their high content of arginine or lysine residues [16]. CPPs are classified into three main categories based on their physicochemical characteristics: amphipathic, cationic, and hydrophobic [17]. These molecules have a very diverse origin. Some are involved in signaling, some are derived from viral proteins, some are part of the antimicrobial defense of various organisms, or they have been designed and screened rationally, aided by computational and/or experimental techniques to form large libraries [17]. Although their structures vary considerably, CPPs share the ability to translocate cellular membranes and successfully release bioactive molecules intracellularly, helping escape endosomes while maintaining high biocompatibility levels [18].

The novel coronavirus SARS-CoV-2 has attracted significant attention over the past two years, not only because of the current global health emergency—where more than 292 million people have been infected, and over 5 million people have died—but the accelerated pace at which biochemical and biophysical studies have been conducted to elucidate the infection mechanisms [19]. The collected data has been fundamental for developing several vaccines in record time [20]. For instance, the spike (S) glycoprotein information indicates that it is the primary source of the viral tropism towards human cells [21]. This protein has a 180 kDa molecular weight, and it is displayed at the viral surface as a trimer composed of two major domains [20]. The first is the S1, which contains the receptor-binding domain (RBD) responsible for mediating the ACE2 (angiotensin-converting enzyme 2) receptor binding. The second one is the S2, which allows membrane fusion through the exposure of a fusion protein activated by proteolytic cleavage in a site upstream (S2′) and proteolytically primed at the interface of the S1 and S2 domains. Transmission of the genetic material into the host cells has been attributed to proteases responsible for priming, receptor binding, and ionic interactions controlling the virus’s stability [20].

By recognizing the strong interaction between the S glycoprotein from the SARS-CoV-2 and the ACE2 of the lung cells membrane, we hypothesized that it was possible to find motifs capable of intermingling with membranes phospholipids and potentially translocate them. This with the ultimate goal of finding much more potent carriers of bioactive molecules to treat various diseases. However, extracting structural information from such complex systems might be challenging [22]. On the one hand, obtaining the three-dimensional structure of selected sequences is rather tricky, considering that one of the protein’s domains is transmembrane [23]. On the other hand, it has been known that the peptides might suffer structural changes along their functional cycle, so precise data about such changes is necessary to accurately extract conformational information that is related to their biological activity [22,24]. Additionally, due to the complexity of lipid bilayers, determining the interactions present at a molecular level experimentally is a daunting task, which is why it is much more convenient to carry out such studies with the aid of computational simulations. In particular, molecular dynamics (MD) simulations have been well-suited for predicting relevant interactions between biomolecules [25].

MD simulations work at an atomic level for systems composed of numerous molecules by describing the atom’s energy based on their positions within the system as a function of time [26], which are, in turn, predicted by a numerical solution of Newton’s equations of motion [27]. These simulations aim to go beyond the basic understanding of intermolecular interactions, guiding new experimental strategies, and they can also be used to explain in detail controversial information obtained at a microscopic or macroscopic scale experimentally [28]. The force fields describing the energy of a system based on the coordinates of its particles [29] have been typically modeled by the following expression (Equation (1)):(1)$$\begin{array}{cc} V\left(r\right)=\underset{\mathrm{bonds}}{\sum }\frac{k_{d}}{2} & {\left(d-d_{0}\right)}^{2}+\underset{\mathrm{angles}}{\sum }k_{\theta }{\left(\theta -\theta_{0}\right)}^{2}\\ & +\underset{\mathrm{dihedrals}}{\sum }\frac{k_{\phi }}{2}\left(1+\cos\left(n\phi -\phi_{0}\right)\right)\\ & +\underset{\mathrm{impropers}}{\sum }\frac{k_{\psi }}{2}{\left(\psi -\psi_{0}\right)}^{2}+\underset{\mathrm{non}-\mathrm{bondedpairs}\left(i,j\right)}{\sum }4\varepsilon_{\mathrm{ij}}\left[{\left(\frac{\sigma_{\mathrm{ij}}}{r_{\mathrm{ij}}}\right)}^{12}-{\left(\frac{\sigma_{\mathrm{ij}}}{r_{\mathrm{ij}}}\right)}^{6}\right]+\underset{\mathrm{non}-\mathrm{bondedpairs}\left(i,j\right)}{\sum }\frac{q_{i}q_{j}}{\varepsilon_{D}r_{\mathrm{ij}}} \end{array}$$

Equation (1). Representation of a typical force field applied to classical biological molecules. The first four terms represent intramolecular contributions to the total energy. The first term corresponds to the bond stretching, k_d_ represents the force constant of the bond, d − d_0_ represents the distance from equilibrium that the atom has moved. Second term is related to deformation angles, k_θ_ is the angle force constant and θ − θ_0_ the angle from equilibrium between 3 bonded angles. Third term is for torsional, where k is dihedral force constant, n the multiplicity of the function, φ the dihedral angle and φ_0_ the phase shift. Fourth term accounts for a planarity term where k_ψ_ is the respective force constant and ψ − ψ_0_ the distance between the 1,3 atoms in the harmonic potential. Fifth and sixth terms refer to the non-bonded interactions known respectively as the Lennard Jones potential and Coulomb [30]. 

Such equations and parameters have been adapted for proper use with lipids and proteins [31] and under different resolution levels that range from low (coarse-grained) to high (quantum-mechanical) depending on the necessary level of accuracy as well as on the desired simulation time scales [32]. For instance, MD has been previously used to predict the structure of lipids surrounding aquaporin-0 (AQP0) [33] and to evaluate the penetration of C_60_ fullerenes into a DSPC membrane model for drug delivery [34].

In this work, the open-source software GROMACS^®^ (Royal Institute of Technology, Uppsala University, Uppsala, Sweden) version 2019.3 [35] was employed for MD simulations of three peptide sequences extracted from the SARS-CoV-2 Spike (S) glycoprotein. The peptides’ ability to translocate eukaryotic lipid bilayers was assessed using a DPPC membrane model under the semi atomistic force field GROMOS93 53a6 by evaluating the Root Mean Square Deviation (RMSD), the radius of gyration (Rg), the z-density profile, and the Potential Mean Force (PMF). Also, we conducted flat-bottom pulling simulations to gain insights into the affinity and dynamics of the interaction of the peptides with the membrane.

Due to the marked tendency of peptides to degrade under typical physiological conditions, we immobilized them on chitosan nanoparticles (CNPs), which have been reported to be biodegradable and biocompatible and, consequently, might be suitable for drug delivery [36,37]. Upon immobilization, we evaluated the biocompatibility of CNPs-peptide nanobioconjugates in vitro through platelet aggregation, hemolysis, and cytotoxicity assays on Vero cells. Also, we conducted cell internalization studies and endosomal escape analyses aided by confocal imaging of the colocalization between the delivered CNPs- peptide nanobioconjugates and Lysotracker^®^ Green.

## 2. Materials and Methods

### 2.1. Selection of Peptides and Structural Analysis

Structural analysis of the S glycoprotein of the SARS-CoV-2 started by extracting the amino acid sequence from the NCBI database (Accession QHD43416.1) [38]. Based on the role in the infection process of host cells and their contribution to cell membrane interaction and fusion, we determined each of the constituent regions of relevance and selected peptide sequences with potential membrane-activity. Once the sequences of interest were chosen, their physicochemical properties were determined, including the GRAVY (grand average of hydropathy) index, net charge, hydrophobicity profile, and molecular weight. The hydrophobicity was particularly important as it largely determined the peptides’ suitability for intermingling with the phospholipids of cell membranes [39]. All the methodological procedure is summarized in Figure 1.

### 2.2. Prediction of Peptides Structure

The prediction of selected peptides’ secondary and tertiary structure was carried out with the amino acid sequences in FASTA format via the i-Tasser server [40,41,42] to initially assess the potential biological activity resulting upon folding [43]. The server generated the top 5 predicted de novo structures in PDB format according to a C-score based on the threading of the template alignments and convergence parameters. The option with the highest C-score, representing the most accurate prediction, was selected for further studies. For comparison, we included a set of three already reported CPP sequences with similar length, molecular weight, charge, and three-dimensional structure. Such CPPs were Buforin II (BUF-II) [44], Frenatin 2.3s [45], and TP2 [46].

### 2.3. Molecular Dynamics Simulations

MD simulations were carried out using the GROMACS^®^ version 2019.3 software with the semi-atomistic Force Field GROMOS96 53a6, which was modified for working with lipid membranes by adding the Berger lipid parameters [47]. A leap-frog integrator was implemented in all simulations, and the integration time steps *δt* was 1 fs. Van der Waals and short-range electrostatic interactions cutoff were set at 1.2 nm, while long-range electrostatics were calculated by the Particle Mesh Ewald (PME) method. Finally, 3-D periodic boundary conditions were imposed on the system.

#### 2.3.1. Behavior of Peptides Inside the Membrane

A simulation box was built consisting of a simplified eukaryotic cell membrane model composed of 128 phospholipid molecules of dipalmitoylphosphatidylcholine (DPPC) and water as a solvent. The evaluated peptides were located vertically at the membrane’s center of mass (COM). Na^+^ or Cl^−^ ions were added to maintain neutrality. Subsequently, energy minimization was carried out to avoid steric hindrance issues, which led to a system with relaxed low-energy conformations [48], then an equilibration of 50,000 steps was carried out at a constant temperature (323 K) using the modified Berendsen thermostat (V-rescale). Lastly, constant pressure (1 bar) of 500,000 steps was imposed by employing the Parrinello–Rahman barostat, ensuring equal conditions in each of the systems’ components. Once the system was correctly parameterized, position constraints were removed, allowing it to interact for 100 ns. The obtained trajectories were further processed to calculate RMSD, radius of gyration, average mass densities, and interaction energies.

#### 2.3.2. Non-Equilibrium Pulling

The peptides were located parallel to the membrane at 6 nm from the bilayer’s headgroups for Umbrella Sampling on a simulation box of 13 nm long and at 5 nm for flat-bottom in a box 12 nm long. The system was solvated with SPC (simple point charge) water model [49], and counterions were added to assure charge neutrality (Figure 2). An NVT equilibration of 50,000 steps at 323 K was run, followed by an NPT equilibration of 50,000 steps at 323 K and 1 bar.

The free energy of the peptides through the lipid membrane was obtained from the Potential Mean Force (PMF) curve built with the data collected from the Umbrella Sampling simulations. To accomplish this, right after the equilibration steps, a 65,000-step steered MD was carried out to transfer the peptide from the aqueous phase into the membrane under a harmonic potential of 600 kJ/mol-nm^2^. Configurations with an average distance of 0.2 nm between them were obtained. Finally, each obtained configuration was taken as an independent simulation, balanced, and minimized again, followed by a production run of 5,000,000 steps. Finally, the PMF profile was obtained, aided by the Weighted Histogram Analysis Method (WHAM) [50].

A deeper understanding of the system’s dynamics was attained by applying a flat-bottom potential of 2000 kJ/mol-nm^2^ at 3.5 nm from the center of mass of the membrane through a simulation run for 400 ns, 200,000,000 steps and a time step of 2 fs. Most suitable peptides for drug delivery obtained through MD simulations were synthesized by GL Biochem Shanghai (Shanghai, China) and subsequently immobilized on chitosan nanoparticles (CNPs).

### 2.4. Synthesis of Low Molecular Weight Chitosan Nanoparticles (CNPs)

CNPs were synthesized by the ionic gelation method [51]. Briefly, 2.4 mg/mL of LMW Chitosan (50–190 kDa, deacetylation degree of 75–85%, CAS 9012-76-4, Sigma-Aldrich, St. Louis, MO, USA) was dissolved in acetic acid 2% *v/v* and left under magnetic stirring for 3 h to protonate the amine groups of monomers and consequently increase its solubility. The pH of the mixture was adjusted to 3.6 to induce a partial charge restoration. Chitosan chains were crosslinked with 1.2 μL of glutaraldehyde per milliliter of chitosan that was added dropwise and was left under stirring for 1 h to obtain the nanoparticles. The obtained CNPs were purified by dialyzing the reaction mixture against Type II water employing a 2 kDa membrane (Sigma-Aldrich, St. Louis, MO, USA) at room temperature for three days. Finally, the CNPs were lyophilized and stored at 4 °C until further use.

### 2.5. Functionalization of CNPs with the Peptides and Rhodamine B

100 mg of CNPs were resuspended in 70 mL of type II water, mixed with 2 mL of glutaraldehyde 2% *v/v*, and left to react for 1 h. 1 mg of the peptide was then added and left to conjugate under continuous agitation for two days. To activate the fluorescent molecule rhodamine B, 7 mg of EDC and 5 mg of NHS were mixed in type II water (5 mL), followed by 200 µL of DMF and 6 mg of rhodamine B. The mixture was left to react at 40 °C for 15 min. Finally, activated rhodamine B was mixed with the CNPs-peptide nanobioconjugates and left under agitation for one day at room temperature. To remove excess rhodamine B, the mixture was dialyzed against Type II water aided by a 2 kDa membrane (Sigma-Aldrich, St. Louis, MO, USA). The labeled nanobioconjugates were lyophilized and stored at 4 °C until further use.

### 2.6. Hemolysis

To evaluate the possible hemolytic tendency of the CNPs-peptide nanobioconjugates, 3 × 10^7^ erythrocytes were collected from a healthy donor in heparin tubes and centrifuged at 1800 RPM for 15 min. The supernatant, corresponding to blood plasma, was removed and replaced with 0.9% *w/v* NaCl solution. The erythrocytes were washed five times with the NaCl solution and then resuspended in PBS 1X. The nanobioconjugates in concentrations ranging from 100 µg/mL to 12.5 µg/mL were diluted in PBS 1X and mixed in triplicate with 100 µL of the erythrocytes into a 96-well microplate and incubated at 37 °C for 1 h. The same procedure was repeated with Triton X-100, which served as the positive control, while PBS 1X was the negative one. After incubation, the microplate was centrifuged at 1800 RPM for 5 min, and the supernatant’s absorbance was read at 450 nm in a microplate spectrophotometer (Thermo Scientific^TM^, Waltham, MA, USA).

### 2.7. Platelet Aggregation

The platelet aggregation capacity of the CNPs- peptide nanobioconjugates was evaluated using platelet-rich plasma (PRP), which was withdrawn from a healthy donor, collected in sodium citrate tubes, and centrifuged at 1000 RPM for 20 min. The nanobioconjugates were suspended in PBS 1X at concentrations ranging from 100 µg/mL to 12.5 µg/mL, mixed with 50 µL of PRP, and poured in triplicate into a 96-well microplate. Thrombin was used as the positive control and PBS 1X buffer as the negative one. Aggregation was estimated 5 min after exposure by reading the absorbance at 620 nm in a microplate spectrophotometer (Thermo Scientific^TM^, Waltham, MA, USA).

### 2.8. Cytotoxicity and Cell Viability

Cytotoxicity of peptides and CNPs-peptide nanobioconjugates was evaluated on Vero cells (ATCC^®^CCL-81) by measuring the metabolic activity associated with the conversion of 3-[4,5-dimethylthiazol-2-yl]-2,5-diphenyltetrazolium bromide (MTT, Sigma-Aldrich, St. Louis, MO, USA) to formazan. 10,000 cells (DMEM, 10% FBS, 1% P/S, Gibco, Amarillo, TX, USA) were transferred to a 96-well microplate and incubated at 37 °C and 5% CO_2_ for 24 h. Culture media was removed from the wells and replaced by the peptide and nanobioconjugate samples. Free peptides and CNPs-peptide nanobioconjugates were added in concentrations ranging from 100 µg/mL to 6.25 µg/mL. Cells grown in DMEM media (supplemented with 1% P/S) were used as negative control while Triton X-100 (Sigma-Aldrich, St. Louis, MO, USA) 1% *v*/*v* was the positive one. Acute and chronic cytotoxicity was measured after exposure to treatments at 37 °C and 5% CO_2_ for 24 and 48 h. Then, 10 µL of MTT was added and was left to react for 2 h before replacing the culture media with 100 µL of DMSO to dissolve the formed formazan crystal. Absorbance was measured at 595 nm in a microplate reader (Thermo Scientific, Waltham, MA, USA).

### 2.9. Plasma Membrane Translocation and Endosomal Escape

Translocation capacity and endosomal escape abilities of bare CNPs and CNPs-peptide nanobioconjugates were assessed by analyzing cellular surface area coverage and colocalization with Lysotracker Green^®^ (Thermo Fisher, Waltham, MA, USA), following internalization in Vero cells (ATCC^®^ CCL-81). To this end, a glass slide was coated with Poly-D-Lysine on which 100,000 cells per well were seeded. Cells were maintained in DMEM medium (5% FBS) at 37 °C and 5% CO_2_ for 24 h to allow cell adhesion. Then, cells were exposed to the nanobioconjugates in an unsupplemented medium at a 25 µg/mL concentration for 30 min and 4 h. Cells were exposed to Hoechst 33342 (1:10,000 with respect to DMEM medium) and Lysotracker Green DND-26 (1:10,000 with respect to DMEM medium) for 5 min and observed by confocal microscopy. The images were acquired in an Olympus FV1000 confocal laser scanning microscope using a PlanApo 60X oil immersion objective. Imaging of nuclei, endosomes, and CNPs-peptide nanobioconjugates was performed at the following excitation/emission wavelengths: 358 nm/461 nm, 488 nm/520 nm, and 546 nm/575 nm, respectively. Finally, the distribution throughout the cytosol was determined by calculating the covered area using the image processing package Fiji (open source). Endosomal escape was estimated by measuring the Pearson correlation coefficient (PCC), which indicates the colocalization of the rhodamine-labeled nanobioconjugates with Lysotracker Green^®^.

## 3. Results and Discussion

### 3.1. Peptides’ Structure Prediction

The S glycoprotein of the coronavirus is 1273 amino acids long. Different subunits have been recognized as playing a specific role when interacting with the ACE2 receptor of cells [52]. The peptides AHB-1, AHB-2, and AHB-3 were selected after identifying their function and main physicochemical properties. AHB-1 was obtained from the signal peptide and is located between residues 1–12 of the N-terminal. The primary function of the signal peptide is to direct the nascent protein to the rough endoplasmic reticulum (RER) membrane [53]. AHB-1 is highly hydrophobic, as indicated by the GRAVY index, and exhibits high stability (Table 1). This was previously reported for highly hydrophobic cell-penetrating peptides derived from other signal peptide sequences [54]. AHB-2 was extracted from the fusion peptide and is located between residues 788–806 of the protein. The fusion peptide has been proved to be released through two proteolytic cleavages during infection and fused into the host’s cell membrane [55]. AHB-2 has an amphipathic and slightly cationic character, which might be advantageous in allowing strong interactions with the membrane model. Finally, AHB-3, derived from the transmembrane region, is located between residues 1214 to 1236 and has high stability due to the alpha-helix folding, promoting membrane permeation [56].

Three more peptides were selected as references, whose translocation capacity has already been studied and proven previously. The first one is Buforin II, an antimicrobial peptide whose activity has been attributed to its ability to enter the intracellular environment and possibly interact with DNA/RNA to interrupt essential survival processes [57]. The second reference was TP2, a potent spontaneous cell translocating peptide that can come across membranes without permanently destabilizing them [46]. The third reference was the antimicrobial peptide Frenatin 2.3s, which has also been reported to have cell translocation abilities [45]. The chosen peptides share common physicochemical properties such as similar molecular size, positive charge, absence of beta-sheets in their secondary structure, and relatively high hydrophobicity.

The peptide’s structure prediction obtained from the de novo modeler i-Tasser server is shown in Figure 3. The obtained confidence levels for the prediction were −0.78 for AHB-1, 0.56 for AHB-2, −0.79 for AHB-3, −1.09 for TP2, −0.65 for Frenatin 2.3s, and 0.14 for Buforin II. C-score intervals range from −5 to 2. Higher values of this score indicate a higher confidence level, showing that the predicted structures are reliable and have the quality required for further molecular dynamics simulations [59].

### 3.2. Molecular Dynamics Simulations

The Root Mean Square Deviation (RMSD) of the peptides’ alpha carbon structure shows that all novel peptides, Frenatin 2.3s and TP2, reached a plateau (with values between 0.1 to 0.3 nm) in about 100 nanoseconds, indicating a regime of conformational stability (see Figure 4). Similar results have been attributed previously to highly stable structures with no significant changes with respect to the initial positions [61]. In contrast, Buforin II exhibited the highest RMSD, reaching up to 0.6 nm; however, no complete stabilization was achieved for the simulation time allotted. This can be explained by some unfolding processes that can span timescales ranging from nanoseconds to milliseconds [62]. 

The 100 ns stability time frame served as a starting point for the simulations involving the phospholipid bilayer and suggested that longer times might destabilize the alpha-helical structures in the presence of the GROMOS96 53A6 force field, as reported by Lemkul et al. [48]. 

The radius of gyration provides insights into the protein’s compactness and tendency to unfold over time in a medium. The compactness of a protein is defined as its surface area compared with a sphere of the same volume [63]. Figure 4 shows that the peptides AHB-1, AHB-2, Frenatin 2.3s, and TP2 maintain high stability throughout the simulation, possibly due to their hydrophobic character, which is highly compatible with the interior of the membrane. AHB-3 showed the lowest Rg, indicating the greatest compactness. However, a significant change was observed between 20 and 40 ns, which can be attributed to a flexible motif in the structure that dynamically undergoes folding and unfolding events over time [64]. Finally, for Buforin II, an increase of 0.5 nm in the Rg was found during the 100 ns of the simulation, indicating that an unfolding process took place and extra time might be needed to reach conformational stability. This could be due to highly hydrophilic residues at the very center of the peptide, which likely induce different interactions with the lipid bilayer’s hydrophobic core.

The membrane was deconstructed into several groups, namely, headgroups, glycerol ester, and acyl chains. The distribution of the membrane’s groups, the peptides, and water was determined in the z-direction, perpendicular to the surface of the bilayer, as represented by the density profiles shown in Figure 5. While AHB-2 remained within the acyl chains, AHB-1 and AHB-3 were evenly distributed throughout the entire membrane and showed stronger interactions with the glycerol ester and the headgroups. The structural profile of the membrane might be altered due to surrounding elements such as water or proteins, which lead to different membrane tensions and corresponding structures [65]. However, no alteration or asymmetry was observed in the profiles obtained along the bilayer. The reference CPPs showed a slight movement of their center of mass towards one side of the membrane, interacting in one side of the headgroups and acyl chains.

The interactions found in the system were divided into bonded and non-bonded. However, non-bonded interactions dominate the molecular scale. Non-bonded interactions are described by the Lennard Jones (LJ) potential and electrostatic (Coulomb) forces [66]. Figure 6A shows that the LJ energy contribution (ranging between 500 to 700 kJ/mol) is similar for all peptides, with slightly higher levels for the novel peptides. As a result of the peptide’s location at the COM of the membrane, the interaction with the hydrophobic core (acyl chains) predominates. Albeit at a lower level, it was observed that the peptides are at a distance in which even the headgroups and the glycerol exert an attractive force.

The headgroups contribute the most to Coulombic interactions because of their partial charges (i.e., negatively charged phosphate and positively charged choline groups). As a result, peptides rich in residues such as arginine and lysine like Buforin II (net charge +6) exhibited a considerable interaction of 1450 kJ/mol. Even though AHB-2 and Frenatin 2.3s have the same charge due to the presence of a lysine residue in their structure, the residue’s position plays a major role in the strength of interaction as evidenced by values of about 1000 and 600 kJ/mol. AHB-1 and AHB-3 are neutrally charged, and consequently, their interaction energies approached only about 320 kJ/mol.

The Potential Mean Force (PMF) obtained from the Umbrella Sampling simulations allowed us to determine the energy landscape quantitatively as the peptide was inserted across the phospholipid bilayer (see Figure 7). The overall shape of the PMF profiles for all the peptides is similar. Peptides were located far enough from the bilayer into the solvent’s bulk to initially neglect any interactions with the phospholipids. Then, at around 2.3 nm, there is an energy minimum located at the solvent-membrane interface, representing the interaction of the peptide with it. The PMF curve then shows a significant energy barrier that indicates the energy expenditure required for the peptide to come across the bilayer. The curve reaches a maximum at about the bilayer center (0 nm).

Free energies (∆*G*) obtained for AHB-1 and AHB-3 were 14.1 and 13.2 kcal/mol, respectively, while for AHB-2, it was 34.9 kcal/mol. Given that the ∆*G* values of Buforin II and TP2 were 8.9 and 23.4 kcal/mol, respectively, it is very likely that AHB-1 and AHB-3 can be considered strong cell penetrators. This could be explained by their high hydrophobicity and neutral net charge that minimizes the interaction with the phospholipid head groups. In contrast, AHB-2, Frenatin, and TP2 exhibit a higher energy expenditure inside the membrane, most likely due to their high affinity with it, favoring them, maintaining a very stable that remains largely unperturbed during the sampling. These results agree well with those reported previously by Yesylevskyy et al., who have also found energy requirements between approximately 16.7 and 47.8 kcal/mol for Penetratin, and TAP peptides to come across a DPPC model membrane [67] and 4.5 kcal/mol for smaller molecules such as Bisphenol A through DPPC according to Chen et al. [68].

Despite the attractive results, it is crucial to remember that Umbrella Sampling entails uncertainty related to the harmonic potential employed to pull the peptide into the membrane. A better understanding of the system’s dynamics was achieved through another non-equilibrium pulling simulation known as flat-bottom. This approach applied a potential if the peptide moved above the position from which the potential was set. The simulation aimed to allow the peptide to interact spontaneously with the membrane by pulling it from a far distance. As a result, the time it took for the peptide to find its most favorable conformation at the bilayer-water interface decreased significantly.

The simulations included a potential imposed and initially applied a force that decreased and stopped pulling the peptides before 0.04 ns and remained inactive along the remaining 400 ns. During the production run, a first stage was observed before 30 ns, where the peptides showed a very dynamic motion profile but remained far from the membrane (Figure 8). At 200 ns, the peptides were almost at the same COM distance, with an orientation that was no longer horizontal, and showed overlapping with the headgroups of the membrane. Finally, at 400 ns, AHB-1 penetrated the membrane almost completely. In contrast, AHB-2 and AHB-3 and the peptides used for comparison only partially penetrated the membrane.

At first, AHB-3 was considered the most promising candidate due to its high stability inside the membrane and the lower energy requirement for translocating it compared with the two other new sequences. However, non-equilibrium pulling indicates that its penetration appears difficult, most likely due to its important affinity with the bilayer. This can be seen in Figure 8, where penetration is not observable after 150 ns. A similar situation was also observed for AHB-2, which has been reported previously by Schaefer et al. [69]. Even though AHB-1 and AHB-3 present similar physicochemical characteristics and behaved similarly in previous simulations, AHB-1 penetrated more into the membrane. This behavior can be explained by its amphipathic structure, which comprises a small hydrophilic head on the C-terminal that interacts strongly with headgroups and a hydrophobic tail that allows it to intermingle with the phospholipid core.

Besides its remarkable ability for membrane penetration, AHB-1 proved to maintain its structural stability during the translocation process. The obtained results strongly suggest that a high cell-penetration capacity is related to positive values for the GRAVY index, which, in turn, correlates well with the presence of hydrophobic residues. Additionally, neutral peptides with only a partially charged motif such as AHB-1 will be more likely to translocate neutral Zwitterionic bilayers such as those found in mammalian cells. In contrast, cationic peptides may be able to translocate more easily bacterial membranes, which are structurally different and negatively charged [70,71]. This contrasts with cationic peptides that penetrate eukaryotic cell membranes by different mechanisms that involve arginine and lysine residues, promoting strong electrostatic interactions with the head groups to form agglomerates that facilitate translocation through pore fusion [72].

To confirm the results obtained in silico, AHB-1 and AHB-2 peptides were synthesized. Also, to extend their lifetime and stability under typical physiological conditions, they were conjugated to Low Molecular Weight (LMW) Chitosan Nanoparticles (CNPs) mainly due to their superior biocompatibility and degradability, and low immunogenicity [73]. Therefore, the obtained CNPs-peptide nanobioconjugates were tested in vitro according to the biocompatibility international standard ISO 10993 that evaluates platelet aggregation, hemolysis, and cytotoxicity in Vero cells. Additionally, the internalization and endosomal escape ability of the nanobioconjugates were also tested in Vero cells.

Hemocompatibility tests were performed considering that the route of administration of these peptides could potentially be intravenous. Peptides with concentrations of up to 300 μg/mL showed negligible hemolytic activity (Figure 9), a value comparable to that of the negative control (i.e., PBS-1X). Similar results were found for platelet aggregation, where the nanobioconjugates exhibited thrombogenic activities comparable to the negative control (PBS-1X), i.e., they remained at around 50% for concentrations up to 150 μg/mL. Similar hemocompatibility results were also reported by us recently for BUF-II-CNPs nanobioconjugates [74]. However, in a situation where the CNPs-AHB-1 nanobioconjugates were to be employed at concentrations above 300 μg/mL, they should be evaluated further, considering that they start to induce mild platelet activation at such concentration levels and, consequently, the possibility of thrombotic complications [75].

The cytotoxic activity of both nanobioconjugates and free peptides were tested for 24 and 48 h in Vero cells by the MTT assay. After exposure to the treatments, cell viability was above 75% after 48 h for concentrations ranging from 6.25 μg/mL to 100 μg/mL (Figure 10). AHB-2 was found to be more cytotoxic than AHB-1, as evidenced by viabilities of 94.03 ± 1.20% for AHB-1 alone and 93.62 ± 1.45% after conjugation compared to 81.73 ± 2.86% and 86.33 ± 1.05% for AHB-2 after 48 h of exposure at the maximum evaluated concentration. Conjugation of AHB-2 to CNPs led to a reduction in cytotoxicity. According to the ISO standard, free peptides and nanobioconjugates can be classified as non-cytotoxic against Vero cells.

Internalization and endosomal escape were evaluated in Vero cells after 30 min and 4 h of exposure (Figure 11 and Figure 12). The percentage of the covered area showed that after 30 min, bare CNPs had internalized more than the nanobioconjugates, with AHB-1-CNPs covering only 29.02 ± 4.12% and AHB-2-CNPs about 34.29 ± 3.10% of the cytosol area. However, after 4 h, CNPs reached a covered cytosol area of 50.95 ± 13.80%, AHB-1-CNPs 83.55 ± 9.69%, and AHB-2-CNPs 44.52 ± 2.33%. These results indicate that the internalization of AHB-2-CNPs was the lowest compared to the AHB-1-CNPs nanobioconjugates and bare CNPs. Remarkably, the AHB-1-CNPs nanobioconjugates increased their coverage by 55.64% from 30 min to 4 h of exposure.

The Pearson’s correlation coefficient (PCC) allows quantifying the colocalization of two fluorophores in an image. The PCC ranges from −1 to 1, where 1 indicates a perfect correlation, i.e., complete colocalization, −1 a perfect inverse correlation, and 0 indicates no correlation, i.e., no colocalization [76]. Figure 11 shows that after 30 min of exposure, there is no statistically significant difference between the colocalization of bare CNPs and CNPs-peptide nanobioconjugates with a PCC that approached 0.73. However, while for the CNPs, the PCC after 4 h remained at the same level, that of the nanobioconjugates decreased significantly, which indicates that they can escape the endocytic internalization pathway to reach the cytoplasm. Because the covered area after 30 min approached 40% with a relatively high PCC, it is very likely that multiple internalization mechanisms may co-occur. In the case of bare CNPs, this notion is supported by the fact that they failed to escape endosomes but increased coverage in time. Therefore, future work will be dedicated to elucidating the mechanistic intricacies of cell penetration.

The obtained results for nanobioconjugates suggest that cell internalization and endosomal escape take place by different routes and mechanisms. This hypothesis is supported by the fact that CPPs confer additional properties to nanoparticles associated with their innate internalization and endosomal escape abilities [77]. Intrinsic characteristics such as charge distribution, length, and structure, as well as additional factors such as peptide concentration, cell type and media properties, directly affect the way in which CPPs interact with cell membranes, leading to different cell-internalization and endosomal escape routes [77,78]. Particularly, cell internalization has been reported to occur by two main routes, energy-dependent endocytosis and energy-independent direct translocation. Based on the obtained results, we suggest that the improved internalization properties of nanobioconjugates could be explained by potential direct-translocation abilities of AHB-1 and AHB-2. This additional internalization route has proved to increase the internalization rates in several nanoparticles conjugated with CPPs such as TAT, Angiopep-2 [77], and Buforin II [79]. Direct translocation abilities have also been reported for Pep-1, MPG, R8, R12, and HIV TAT peptides [78] through different mechanisms such as pore formation, inverted micelle, and carpet model [78].

On the other hand, several works have reported the ability of CPPs to induce en-dosomal escape. This ability is manly associated with the capacity of CPPs to interact with endosomal membranes, inducing membrane disruption via pore formation and leakage or by generating ionic pairs with negatively charged membrane lipids [78]. The obtained results confirmed the ability of AHB-1 and AHB-2 to improve the endosomal escape properties of CNPs, leading to a better performance as delivery vehicles. However, it is well known that endosomal escape is a significant limitation in the design of nanovehicles based on CPPs conjugation [77]. Therefore, it is crucial to implement additional strategies to successfully solve this issue. For this, we propose the conjugation of different molecules such as pH-responsive polymers, endosomolytic agents, fusogenic lipids, or photosensitizers to improve endosomal escape via proton sponge effect, membrane disruption, membrane fusion or photochemical internalization, respectively [77,78]. Future studies will focus on determining cell-internalization routes through implementation of routes inhibitors as well as functional experiments to determine and quantify endosomal escape. In addition, high-performance vehicles will be developed by combining the proposed nanobioconjugates with further molecules to improve endosomal escape.

Results also lead to the conclusion that the endosome escape of the nanobioconjugates is a relatively slow process that requires several hours. Similar findings have been reported previously, where it was demonstrated that internalization of functionalized CNPs mainly proceeded by endocytosis, followed by their localization in the cytoplasm up to 24 h post-exposure [80]. Importantly, molecular dynamics simulations made an accurate prediction about the stability and affinity of the AHB-1 peptide with the DPPC lipid membrane model, which showed promising results validated by in vitro experiments. We propose to continue with AHB-3 testing because, according to MD simulations, it showed a behavior that is similar to AHB-1 and therefore it might also exhibit a superior cell-penetration potency.

## 4. Conclusions

One of the major challenges of modern pharmacology is to assure that administered drugs not only reach the site of action but internalize the target cells efficiently. A promising route to achieve this is with the aid of cell-penetrating biomolecules and particularly peptides. We hypothesized that the spike glycoprotein of the SARS-CoV-2 was a suitable source of such sequences. In this context, we identified three potential new sequences (named AHB-1, AHB-2, and AHB-3) and tested them in silico and in vitro in search of superior internalization and endosome escape capacities while avoiding a significant reduction in cell viability.

In this regard, MD simulations of the peptides interacting with model membranes allowed us to determine that out of the three identified sequences, AHB-1 showed superior capacity for intermingling with phospholipid bilayers without either disrupting them or altering their original conformation. In vitro assays of the peptides conjugated to chitosan nanoparticles (peptide-CNPs nanobioconjugates) provided further evidence for this notion and additionally demonstrated high biocompatibility, as evidenced by high biocompatibility in terms of cell viability (in Vero cells), and hemolytic (below 1%) and platelet aggregation tendencies (about 50%). Moreover, it was found that the AHB-1-CNPs nanobioconjugates reached about 90% cytosol coverage compared with about 45% for AHB-1-CNPs nanobioconjugates. Finally, for both nanobioconjugates, endosomal escape approached about 30%, which suggested the interplay of various internalization and intracellular trafficking mechanisms.

Taken together, our results indicate that viral proteins appear a suitable source of novel CPPs and that despite marked differences in timescales with respect to experimental data, MD simulations provide useful insights into the mechanisms underlying peptide-bilayer interactions. Future work will be dedicated to exploring in detail the internalization mechanisms in various cell lines along with a more comprehensive biological testing involving animal models.

## Figures and Tables

**Figure 1 membranes-12-00600-f001:**
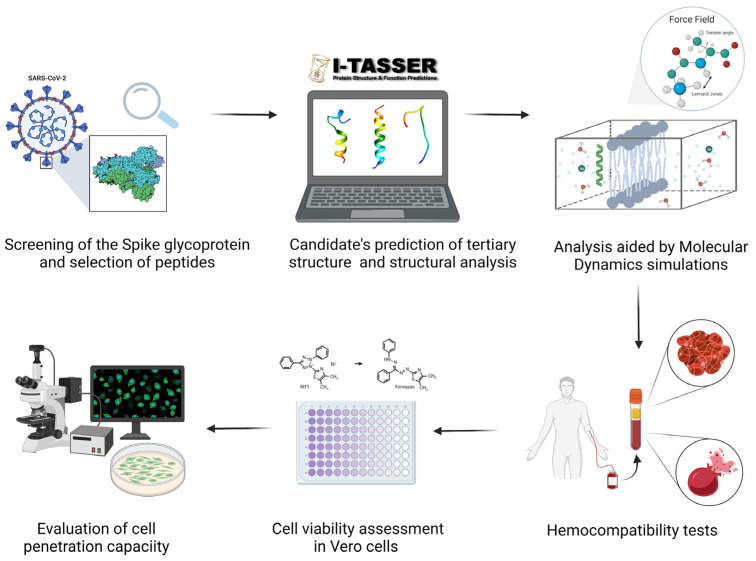
Schematic of the methodology for the rational search of peptides with translocating capacity aided by computational strategies and in vitro testing that included cell internalization and endosome escape, platelet aggregation tendency, hemolytic behavior, and cytotoxicity.

**Figure 2 membranes-12-00600-f002:**
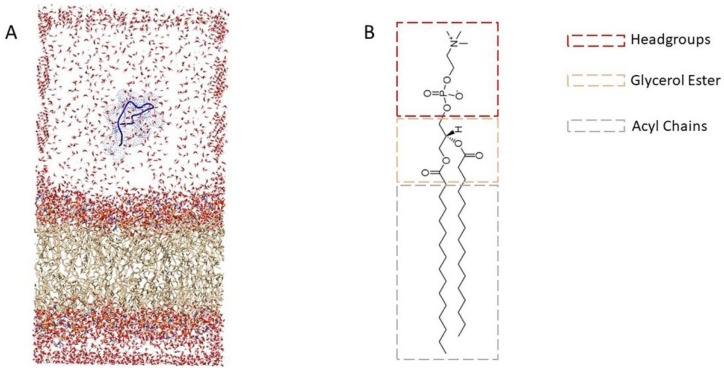
(**A**) Snapshot of the initial configuration of the DPPC membrane in the presence of the AHB-2 peptide, which was in the water at 5 nm from the COM of the membrane. (**B**) Chemical structure of dipalmitoylphosphatidylcholine (DPPC) with its main components indicated by dotted boxes.

**Figure 3 membranes-12-00600-f003:**
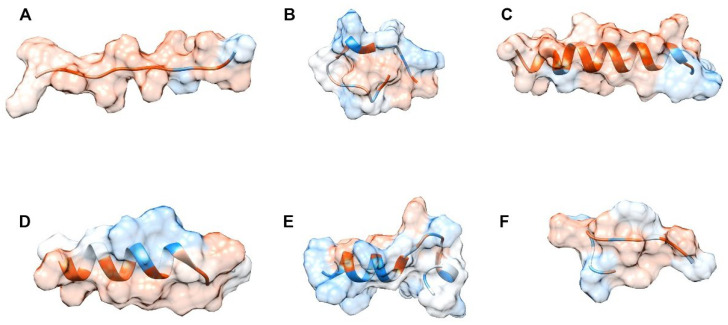
De novo prediction of tertiary structures. Model peptide visualization as predicted by UCSF Chimera^®^, employing color variations based on the residues hydrophobicity score of the Kyte-Doolittle scale [60], where blue is for the most hydrophilic residues and white and orange for the most hydrophobic ones. (**A**–**C**) correspond to the AHB-1, AHB-2, and AHB-3 peptides. (**D**–**F**) correspond to Frenatin 2.3S, Buforin II, and TP2.

**Figure 4 membranes-12-00600-f004:**
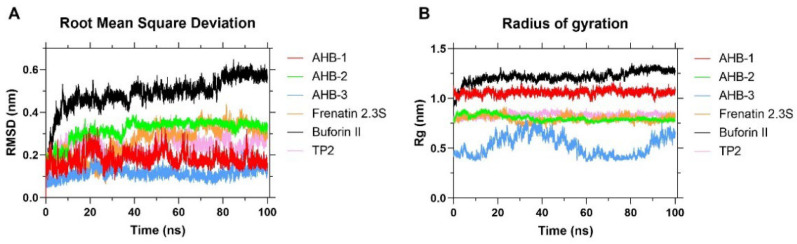
(**A**) Root Mean Square Deviation (RMSD) of the peptides’ carbon alpha structure as a function of time. (**B**) Radius of Gyration (Rg). Each MD simulation lasted for 100 ns.

**Figure 5 membranes-12-00600-f005:**
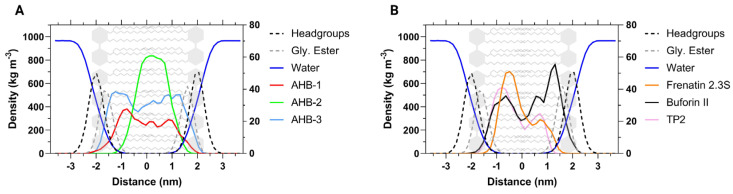
Comparison of average density profiles obtained from a 100 ns simulation. (**A**) novel peptides AHB-1, AHB-2, and AHB-3. (**B**) Reference CPPs Frenatin 2.3s, Buforin II, and TP2. The right *y*-axis presents the density scale for peptides, while the left *y*-axis that of the membrane components and bulk water. Headgroups delimit the thickness of the membrane and the interface with water.

**Figure 6 membranes-12-00600-f006:**
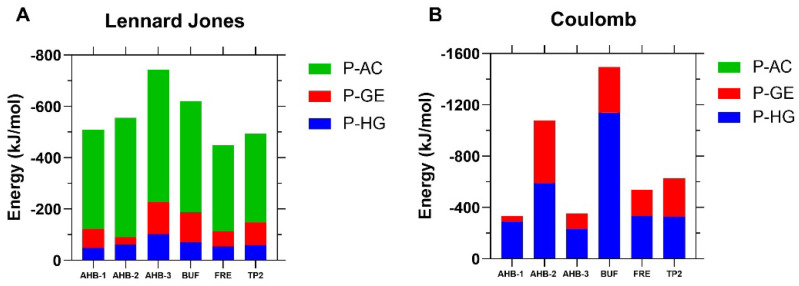
Interaction energies between the peptides and the membrane’s groups. (**A**) Non-bonded energies described by the Lennard Jones Potential and separated into the contribution for each membrane’s components. (**B**) Electrostatic interaction energies. P-AC: Peptide-Acyl Chains interactions, P-GE: Peptide-Glycerol Ester interactions, and P-HG: Peptide-Headgroups interactions.

**Figure 7 membranes-12-00600-f007:**
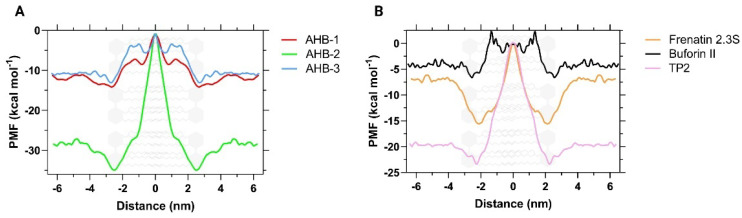
Potential of Mean Force (PMF) calculated by Umbrella Sampling simulations after applying a WHAM analysis. (**A**) Comparison between novel peptides and (**B**) comparison of control CPPs. 0 nm represents the center of mass (COM) along the *z*-axis of the phospholipid bilayer.

**Figure 8 membranes-12-00600-f008:**
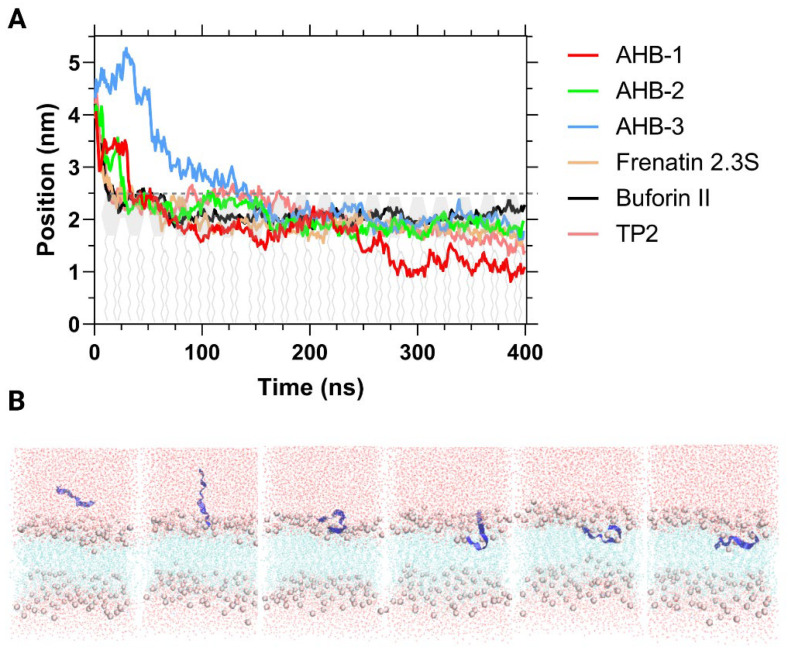
(**A**) Average distance between peptide and membrane’s COM with a flat-bottom restraining potential during 400 ns of simulation. The dotted grey line represents the water-membrane interface. (**B**) Snapshots of AHB-1 peptide trajectory at 25, 70, 120, 200, 250, and 320 ns along the span of the DPPC membrane.

**Figure 9 membranes-12-00600-f009:**
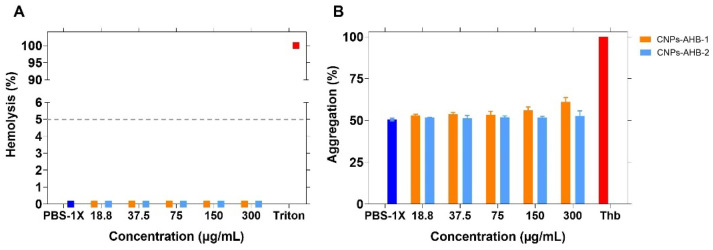
In vitro biocompatibility assays for CNPs-AHB-1 (in orange) and CNPs-AHB-2 (in light blue) nanobioconjugates. (**A**) Nanobioconjugates exhibited a low hemolytic activity comparable to the PBS-1X used as a negative control. The positive control was Triton X-100. (**B**) Nanobioconjugates showed low platelet aggregation compared with PBS-1X as negative control and Thrombin as a positive control.

**Figure 10 membranes-12-00600-f010:**
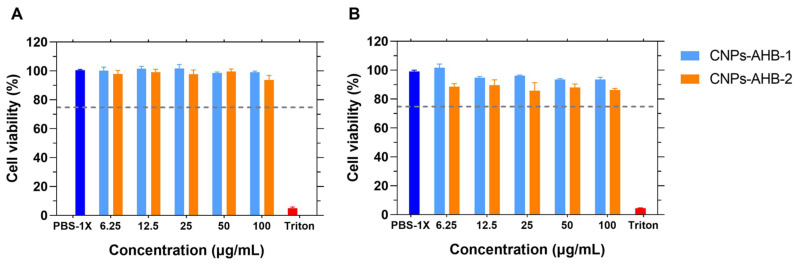
Cell viability of free peptides and CNPs-peptide nanobioconjugates in Vero cells by the MTT assay. (**A**) Percentage of viable cells after 24 h of exposure to the treatments. (**B**) Percentage of viable cells after 48 h of exposure to the treatments. Cell viability remained above 85% when exposed to CNPs-peptide nanobioconjugates. PBS-1X was used as negative control and Triton X-200 0.2% (*v/v*) as positive control.

**Figure 11 membranes-12-00600-f011:**
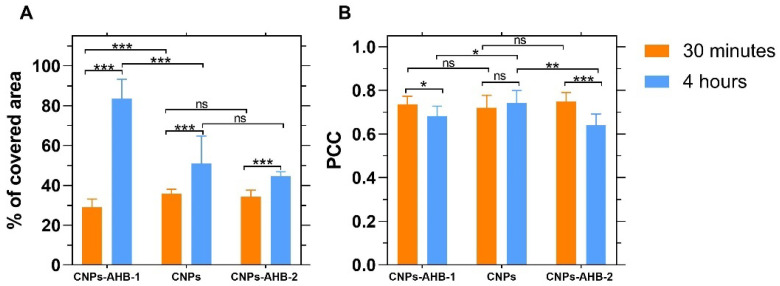
(**A**) Percentage of the covered area by the CNPs in Vero cells. (**B**) Pearson correlation coefficient (PCC) as an indicator of colocalization between CNPs-peptide nanobioconjugates and endosomes labeled with Lysotracker Green^®^. In orange, results obtained after 30 min of exposure and in light blue after 4 h. Asterisks represent *p* values from statistical analysis where * is for *p* ≤ 0.05, ** for *p* ≤ 0.01, and *** for *p* ≤ 0.001.

**Figure 12 membranes-12-00600-f012:**
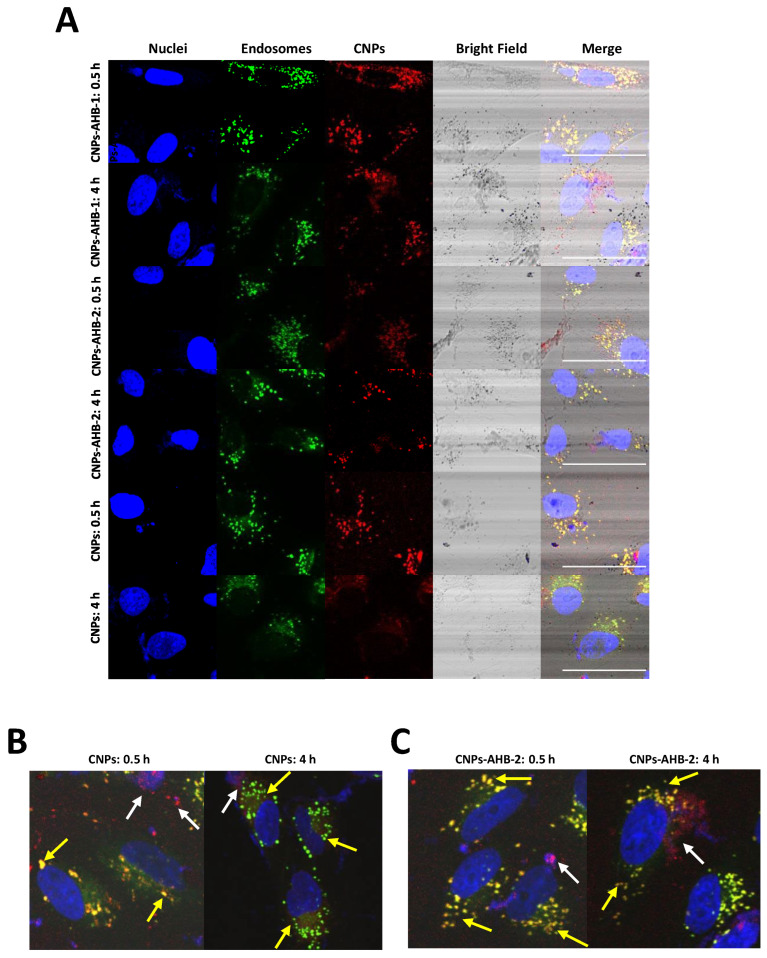
(**A**) Cellular internalization of Vero cells by bare Chitosan nanoparticles (CNPs) and CNPs-AHB-1 and CNPs-AHB-2 nanobioconjugates after 30 min and 4 h of exposure. Imaging was conducted at a 60× magnification. The scale bar corresponds to 100 μm. (**B**) Vero cells imaged at a 60× magnification, and a digital zoom adjusted to 120× after 30 min and 4 h of exposure to CNPs. (**C**) Vero cells imaged at a 60× magnification, and a digital zoom adjusted to 120× after 30 min and 4 h of exposure to CNPs-AHB-2 nanobioconjugates. The yellow arrows point to highly colocalized green and red channels, which indicates entrapment of CNPs or CNPs-AHB-2 nanobioconjugates into endosomal compartments. White arrows indicate regions with a prevalence of the red channel, which correlates with the endosomal escape of CNPs or CNPs-AHB-2 nanobioconjugates.

**Table 1 membranes-12-00600-t001:** Primary sequence and physicochemical properties of evaluated peptides (AHB-1, AHB-2, and AHB-3) extracted from the ProtParam tool by ExPASy [58] and comparison with reference peptides (Frenatin 2.3s, Buforin II, and TP2).

Peptide	AHB-1	AHB-2	AHB-3	Frenatin 2.3s	Buforin II	TP2
Sequence	MFVFLVLLPLVS	IYKTPPIKDFGGFNFSQIL	WYIWLGFIAGLIAIVMVTIMLCC	GLVGTLLGHIGKAILGG	TRSSRAGLQFPVGRVHRLLRK	PLIYLRLLRGQF
Residues	12	19	23	17	21	12
(Asp + Glu)	0	1	0	0	0	0
(Arg + Lys)	0	2	0	1	6	2
Net Charge	0	+1	0	+1	+6	+2
GRAVY	+2.74	+0.03	+2.3	+1.18	−0.64	+0.56
Theo. pI	5.28	8.5	5.51	8.76	12.6	10.84
mW	1377.79	2185.55	2630.36	1575.91	2434.88	1488.84

## Data Availability

No new data were created or analyzed in this study. Data sharing is not applicable to this article.

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
