# Peer review of "Translocating Peptides of Biomedical Interest Obtained from the Spike (S) Glycoprotein of the SARS-CoV-2"

_membranes, 2022, doi:10.3390/membranes12060600_

Round 1

Reviewer 1 Report

I have gone through the manuscript which provides novel information. I recommend its publication in the journal of membranes. 

Author Response

Translocating peptides of biomedical interest obtained from the Spike (S) glycoprotein of the SARS-CoV-2

Journal: Membranes

Response to reviewers:

The authors would like to express their gratitude towards the reviewers and the Editor-in-Chief for their insightful and constructive comments on the manuscript. The original remarks from the reviewers are shown below in bold. Changes in the document are explained herein and highlighted with the aid of the tracking changes tool of Microsoft Word®.

Reviewer 1:

I have gone through the manuscript which provides novel information. I recommend its publication in the journal of membranes.

Response: We appreciate the reviewer´s positive evaluation of our manuscript.

Reviewer 2 Report

See attached file

Author Response

Translocating peptides of biomedical interest obtained from the Spike (S) glycoprotein of the SARS-CoV-2

Journal: Membranes

Response to reviewers:

The authors would like to express their gratitude towards the reviewers and the Editor-in-Chief for their insightful and constructive comments on the manuscript. The original remarks from the reviewers are shown below in bold. Changes in the document are explained herein and highlighted with the aid of the tracking changes tool of Microsoft Word®.

Reviewer 2:

The manuscript “Membranes-1741221” reports a complete computational study and an experimental analysis of the cytotoxicity of a hot-topic, namely the SARS-CoV-2 virus. In particular, the authors have focussed their efforts on the study of the effects of a novel peptide called AHB-1 on the task of membrane permeation of bioactive molecules when attached to chitosan nanoparticles.

The authors have performed a series of all-atom molecular dynamics simulations combined with umbrella sampling (WHAM) calculations and experimental measurements. The methods reported are scientifically correct and also described with enough details.

In summary, the work is solid and sound, with interesting findings and it should be considered for publication in “Membranes”, but only after major revision. In order to improve the manuscript, several remarks should be considered by the authors:

  1. In the “Abstract”, kcal mol-1 should be kcal/mol.

Response: the abstract was adjusted according to the reviewer´s suggestion

  1. In Eq. 1., variables and parameters should be clearly described or referred, i.e. which kind of angles are, what is “d”, etc.

Response: the description of the missing variables were added to the manuscript.

  1. In MD description, temperature and/or pressure are not indicated. Important to know them.

Response: the values for temperature and pressure employed for MD simulations were included.

  1. About Fig 4: radii of gyration look very stable, but the case of ANB-3 seems to perform some periodic motion of about 100 ns, i.e. not stable. Is this true? In my opinion a longer time interval should be performed to explore this fact.

Response: we certainly agree with the reviewer that MD simulations need to be extended to explore the complete mechanism of interaction and elaborate more compelling hypotheses in that regard. As of now, our simulations were only used as an indication of the potential of a peptide to penetrate the membrane.

  1. The background figures in Fig.5 are unclear and they should probably be better highlighted. References of plot B look not symmetrical, indicating 100 ns is rather short for these profiles. Please comment.

Response: The background and references were highlighted and aligned according to the reviewer´s suggestion. As discussed above, 100 ns is probably a short time but sufficient for the main purpose of our studies, which was to screen for sequences with high membrane-penetration capabilities. The lack of symmetry in the density profiles shown in figure 5B is also related to the analysis based on a single peptide. The symmetry would be easily achieved with multiple peptides inside the membrane as the probabilities of approaching either side of the membrane would counterbalance due to strong interactions with both headgroups. Therefore, the presence of various peptides leads to balanced distributions and symmetric profiles. Nevertheless, the single peptide analysis also provides valuable information. Symmetric profiles relate to not so strong interactions that allow back-and-forth peptide displacement between opposite headgroups. In contrast, asymmetric profiles are indications of stronger interactions that shift the concentration towards one of the membrane ends.      

  1. PMFs reported in Fig.7 give large barriers (up to 35 kcal/mol) and since they are significantly large, it would be interesting a comparison with additional works or other similar systems (i.e. large proteins with small-solutes) as a sort of validation, apart of Ref. 65.

Response: as suggested by the reviewer, we added a much more comprehensive discussion of energy barriers for various molecules and biomolecules of interest in the study of membrane translocation.

  1. 12 A quite blurry, can you improve it please? Not clear enough.

Response: the figures qualities were improved to favor readability.

Reviewer 3 Report

Based on the crucial role of SARS-CoV-2 S glycoprotein in host cell tropism by the virus, the authors utilized specific segments of the viral protein to generate novel peptides which would improve bioactive molecules delivery into target cells. Overall, this study has been well-designed and has provided interesting output for other pharmacology studies in the future. However, I have several comments that should be taken into consideration to further improve the significance of the manuscript:

1) As the authors are utilizing SARS-CoV-2 S glycoprotein-based peptides, it is important to take note of the high IFN response induced during a regular SARS-CoV-2 infection. As Vero cells are known to be IFN-deficient cell line, it may not be the best model to evaluate the cytotoxicity properties of the peptides. Any possible cytotoxicity observed here could be underestimated.

2) A positive control for cytotoxicity such as cycloheximide should be used in the assay.

3) It will also be interesting to test the peptides' plasma membrane translocation and endosomal escape properties in primary cells as it is better represents the actual physiological conditions.

4) Apart from the in silico analyses, the authors should compare the novel peptides' cellular internalization properties with other known cell penetrating peptides; or further discuss about this in the discussion.

Author Response

Translocating peptides of biomedical interest obtained from the Spike (S) glycoprotein of the SARS-CoV-2

Journal: Membranes

Response to reviewers:

The authors would like to express their gratitude towards the reviewers and the Editor-in-Chief for their insightful and constructive comments on the manuscript. The original remarks from the reviewers are shown below in bold. Changes in the document are explained herein and highlighted with the aid of the tracking changes tool of Microsoft Word®.

Reviewer 3:

Based on the crucial role of SARS-CoV-2 S glycoprotein in host cell tropism by the virus, the authors utilized specific segments of the viral protein to generate novel peptides which would improve bioactive molecules delivery into target cells. Overall, this study has been well-designed and has provided interesting output for other pharmacology studies in the future. However, I have several comments that should be taken into consideration to further improve the significance of the manuscript:

  1. As the authors are utilizing SARS-CoV-2 S glycoprotein-based peptides, it is important to take note of the high IFN response induced during a regular SARS-CoV-2 infection. As Vero cells are known to be IFN-deficient cell line, it may not be the best model to evaluate the cytotoxicity properties of the peptides. Any possible cytotoxicity observed here could be underestimated.

Response: Vero cells were employed in our studies because they are considered the gold standard for cytotoxicity according to the ISO 10993. We certainly agree with the reviewer that infection might be important for other cell lines; however, our studies focused only on cell-penetration capabilities of small moieties within the spike protein and not on possible infection routes. We appreciate the reviewer for pointing this out as it could be an additional and exploitable source of information for future studies.

  1. A positive control for cytotoxicity such as cycloheximide should be used in the assay.

Response: the positive control of our cytotoxicity studies was Triton X-100, which was used to normalize the absorbance values and calculate what is shown in Figure 10 for all treatments. According to the reviewer´s suggestion, we included both the positive a negative control in such plots.

  1. It will also be interesting to test the peptides' plasma membrane translocation and endosomal escape properties in primary cells as it is better represents the actual physiological conditions.

Response: We agree with the reviewer that it will be quite valuable and informative to evaluate other cell lines’ responses to our peptides. This is indeed the focus of our ongoing experiments and the results will be published in an upcoming contribution.

  1. Apart from the in silico analyses, the authors should compare the novel peptides' cellular internalization properties with other known cell penetrating peptides; or further discuss about this in the discussion.

Response: according to the reviewer´s suggestion, we included a much more detailed description of how our peptides compare with previous experimental results by us and other research groups.

Round 2

Reviewer 2 Report

Dear authors, after the revision, although the simulations have not been extended (I assume this would require long time), I think the paper is ready for publication.

Reviewer 3 Report

The authors have addressed all my concerns and have made the necessary revisions to improve the manuscript.